# Optimization of Thyroid Volume Determination by Stitched 3D-Ultrasound Data Sets in Patients with Structural Thyroid Disease

**DOI:** 10.3390/biomedicines11020381

**Published:** 2023-01-27

**Authors:** Philipp Seifert, Sophie-Luise Ullrich, Christian Kühnel, Falk Gühne, Robert Drescher, Thomas Winkens, Martin Freesmeyer

**Affiliations:** Clinic of Nuclear Medicine, Jena University Hospital, Am Klinikum 1, 07747 Jena, Germany

**Keywords:** thyroid, ultrasound, 3D, volumetric determination, stitching

## Abstract

Ultrasound (US) is the most important imaging method for the assessment of structural disorders of the thyroid. A precise volume determination is relevant for therapy planning and outcome monitoring. However, the accuracy of 2D-US is limited, especially in cases of organ enlargements and deformations. Software-based “stitching” of separately acquired 3D-US data revealed precise volume determination in thyroid phantoms. The purpose of this study is to investigate the feasibility and accuracy of 3D-US stitching in patients with structural thyroid disease. A total of 31 patients from the clinical routine were involved, receiving conventional 2D-US (conUS), sensor-navigated 3D-US (3DsnUS), mechanically-swept 3D-US (3DmsUS), and I-124-PET/CT as reference standard. Regarding 3DsnUS and 3DmsUS, separately acquired 3D-US images (per thyroid lobe) were merged to one comprehensive data set. Subsequently, anatomical correctness of the stitching process was analysed via secondary image fusion with the I-124-PET images. Volumetric determinations were conducted by the ellipsoid model (EM) on conUS and CT, and manually drawn segmental contouring (MC) on 3DsnUS, 3DmsUS, CT, and I-124-PET/CT. Mean volume of the thyroid glands was 44.1 ± 25.8 mL (I-124-PET-MC = reference). Highly significant correlations (all *p* < 0.0001) were observed for conUS-EM (r = 0.892), 3DsnUS-MC (r = 0.988), 3DmsUS-MC (r = 0.978), CT-EM (0.956), and CT-MC (0.986), respectively. The mean volume differences (standard deviations, limits of agreement) in comparison with the reference were −10.50 mL (±11.56 mL, −33.62 to 12.24), −3.74 mL (±3.74 mL, −11.39 to 3.78), and 0.62 mL (±4.79 mL, −8.78 to 10.01) for conUS-EM, 3DsnUS-MC, and 3DmsUS-MC, respectively. Stitched 3D-US data sets of the thyroid enable accurate volumetric determination even in enlarged and deformed organs. The main limitation of high time expenditure may be overcome by artificial intelligence approaches.

## 1. Introduction

Depending on regional circumstances, structural deviations of the thyroid gland such as nodules and organ enlargements frequently occur. In areas of iodine deficiency, such as Germany, parenchymal disorders can be observed in >30% of the population [1,2]. Ultrasound (US) is the most important imaging method for the assessment of thyroid diseases, allowing for detailed morphological estimations as well as organ and lesion size measurements [3,4]. Its widespread availability, almost no side effects, high level of acceptance by both healthcare professionals and patients, excellent resolution and contrast in soft tissue, real-time applicability, portability, possibility of de facto unlimited follow-up scans, and very low procedural costs are well-known advantages of US examinations in general [5,6,7].

An accurate US-based determination of the thyroid volume is relevant for treatment planning, outcome assessment, and follow-up in the therapy of thyroid diseases. Precise volume measurements are especially important in the preparation of radioiodine therapies and minimal-invasive surgical approaches [8,9,10]. However, the diagnostic accuracy of conventional 2D-US (conUS) is limited by intraobserver and interobserver variability, a restricted field of view (FOV), and the invisibility of retrosternal or intrathoracic thyroid tissue [11,12,13].

Some of these limitations can be overcome by 3D-US applications, especially in view of volume determination [14]. The possibility of segmental contouring, in analogy to other sectional imaging methods such as CT or MRI, enables significant improvement of the diagnostic accuracy [15,16]. Furthermore, it is possible to export and archive acquired 3D-US data sets which allow for retrospective reviewing, post-processing, post-hoc volume analyses, and secondary fusion with other 3D imaging modalities [17]. Various technical methods of 3D-US data acquisition have been developed, including sensor-navigated (3DsnUS) and mechanically swept approaches (3DmsUS) [18].

However, 3D-US is limited to the FOV of the used US probe, restricting precise volume measurements especially in case of isthmus enlargement. One possible solution to overcome this limitation is the so-called “stitching” technique, in which two separately acquired 3D-US data sets per thyroid lobe are merged together. In previous phantom studies the feasibility and accuracy of this novel software-based method has been analysed. The results revealed extremely precise volume measurements of the investigated thyroid phantoms even in case of very large and deformed organs [19,20]. The aim of the present study was to translate the stitching technique into the clinic setting and to investigate its feasibility and accuracy for both 3DsnUS and 3DmsUS.

## 2. Materials and Methods

Subject of the study were 31 patients with structural thyroid disease recruited from the clinical routine of a university nuclear medicine between April 2015 and November 2016. When conventional thyroid diagnostics, consisting of history, clinical examination, laboratory parameters, sonography, and scintigraphy, revealed unclear findings, the patients received additional I-124-PET/CT and subsequent PET/US fusion imaging as part of several different clinical study protocols, published in 2019 and 2020 [21,22,23]. Therefore, PET/CT was conducted independently of the present study protocol. In a part of these patients, additional 3D-US investigations were performed. This represents the object of interest of this work.

All patients were informed in detail and signed a consent form. The 3D-US examination results had no influence on the treatment course of the respective thyroid diseases. Local ethics committee approved the scientific evaluation of the collected data (reg. no.: “2021-2376-Daten”).

### 2.1. Ultrasound Examinations

US was performed on the LOGIQ E9 device (GE Medical Systems, Milwaukee, WI, USA). Separate scans of each thyroid lobe (left and right) were acquired. ConUS was conducted with the linear matrix array ML6-15 according to a local standard operating procedure [7]. For 3DsnUS, a magnetic field and specific position sensors equipped to the ML6-15 probe were necessary. For 3DmsUS an automated mechanically swept 3D convex probe (RAB4-8) was used. The methodology of these 3D-US applications has been described in several previous publications [14,19,20]. All 3D-US data sets were transferred to the research software PMOD (Version 4.1, PMOD Technologies Ltd., Zürich, Switzerland). Examination settings and acquired data sets are depictured in Figure 1 and Figure 2.

The parameter settings for the two investigated 3D-US applications are shown in Table 1.

### 2.2. Stitching of 3D-US Data Sets

In both settings, 3DsnUS and 3DmsUS, separately acquired scans (one scan per thyroid lobe: left and right) were merged together (=stitching) in PMOD software by use of a specific stitching module. To stitch both data sets together, it was possible to move one previously defined thyroid lobe on each plane and manually align it according to the assumed anatomy. The tool allowed for translations in each of three axes (x, y, z) as well as rotations within each of the three planes (transverse, sagittal, and coronal). To provide a consistent basis for processing, the larger thyroid lobe was constantly defined as fixed and the smaller as movable. Suitable pivot point for rotary movements were set as centrally as possible in the respective isthmuses. Initial starting point for the stitching procedure was the transverse plane. Various anatomical landmarks were used for orientation. Beforehand, an overview of possible features or anomalies was carried out. Those should be located as close as possible to the isthmus in order to be mapped on both data sets. Parenchymal changes such as nodes or cysts close to the isthmus and calcifications of the trachea proved helpful. These were aligned and overlapped in all axes.

The finished stitched 3D-US data sets could then be transferred to the clinical software Syngo.via (version VB50B, Siemens, Erlangen, Germany). Herein, secondary fusion with the available I-124-PET scans was conducted, allowing a precise assessment of the anatomical correctness of the 3D-US stitching process. According to the findings, respective corrections were performed on PMOD software and again re-evaluated by I-124-PET fusion imaging on Syngo-via software (Figure 3).

### 2.3. Volumetric Determination

Two different methods were used to measure the thyroid volumes. ConUS data were evaluated via the ellipsoid model (conUS-EM): V = (4/3) * π * (largest cranial-caudal diameter/2) * (largest anterior-posterior diameter/2) * (orthograde medial-lateral diameter/2) [24]. Volume determination of the stitched 3D-US data sets were performed by multiple manually drawn segmental contouring applications (MC) in transverse plane in PMOD software according to the organ boarders (3DsnUS-MC, 3DmsUS-MC). The same method was also used to measure the thyroid volumes on I-124-PET/CT scans in Syngo.via software (PET/CT-MC), which defined the reference standard values. To avoid prejudiced biases, determination of the reference was performed only after the measurements of the thyroid volumes on the several US data sets. For further comparative analyses, the CT scans alone were additionally evaluated in Syngo.via software using both the MC and EM methods (CT-EM, CT-MC). ConUS-EM and CT-EM applications are demonstrated in Figure 4.

MC applications on 3D-US (3DsnUS-MC and 3DmsUS-MC), CT (CT-MC), and I-124-PET/CT (PET/CT-MC) are shown in Figure 5.

### 2.4. Statistics

Data were recorded in Excel software (version 16.59, Microsoft Corporation, Redmond, WA, USA). Statistical analysis was performed using SPSS software (version 28.0.1.0, IBM Corporation, Ehningen, Germany). Diagrams and charts were created in Excel. Figures were prepared in PowerPoint software (version 16.62, Microsoft Corporation, Redmond, WA, USA). Bland–Altman plots were created to assess the agreement of the several measured volumes with the reference standard. For the comparison of not normally distributed metric values, Mann–Whitney U test (MWU) was used. For correlation analyses, the Pearson correlation coefficient was applied (r). *p* < 0.05 was considered statistically significant.

## 3. Results

A total number (N) of 31 patients (12 men, 19 women, age: 57 ± 15 years) were included, 21 patients received 3DsnUS and 3DmsUS, 10 patients only received 3DsnUS but no 3DmsUS. Hence, a total of 52 3D-US examinations were carried out. In all cases, parenchymal changes were present in the form of thyroid nodules (multi-nodular goitre in 87.1%). Application of 3D-US, stitching of the separately acquired 3D-US data sets in PMOD software, and secondary image fusion of the stitched comprehensive 3D-US with the I-124-PET image in Syngo.via software was feasible in all cases. Two examples of patients with isthmus nodules are shown in Figure 6.

Cumulative results of all volume determination are displayed in Table 2.

All investigated methods revealed very strong correlations with the reference standard with superior values (not significant) for 3D-US. No significant differences between 3DsnUS-MC and 3DmsUS-MC were observed. A comparative graphical presentation, visualizing the broader spread of the conUS-EM versus 3D-US-MC measurements, is shown in [Fig biomedicines-11-00381-ch001].

Additionally, the Bland–Altman plots demonstrate the wider deviations (in relation to the reference standard PET/CT-MC) of conUS-EM volume measurements in comparison to 3DsnUS-MC and 3DmsUS-MC ([Fig biomedicines-11-00381-ch002]). The mean (SD, LoA) of the measured volume differences were −10.50 mL (±11.56, −33.62 to 12.24), −3.74 mL (±3.74, −11.39 to 3.78), and 0.62 mL (±4.79, −8.78 to 10.01) for conUS-EM, 3DsnUS-MC, and 3DmsUS-MC, respectively.

## 4. Discussion

The present pilot study represents the first in vivo application of stitched 3D-US data sets for volume measurements of the thyroid gland in patients with structural thyroid disease. Stitching of two separately acquired 3D-US data sets has been investigated in two previous phantom studies [19,20]. The volumes determined by the reference standard (multiple manually drawn segmental contouring of I-124-PET/CT scans) revealed a patient cohort with enlarged organs (44.1 ± 25.8 mL). 

The PET/CT scans were not performed for the purpose of this study but as part of other studies [21,22,23]. I-124-PET/CT-MC was considered a very accurate reference standard method due to the hybrid approach including functional (PET) and morphological (CT) information. CT alone has been described as a suitable 3D imaging reference for thyroid volume determination, but the limited soft tissue contrast can lead to inaccurate measurements [14,25,26]. I-124-PET alone, on the other hand, may include areas without iodine uptake (e.g., cysts, calcifications, or hypofunctioning lesion) and, therefore, might underestimate the actual thyroid volume. The combination of both methods has the potential to sufficiently compensate each other’s weaknesses.

Sensor-navigated 3D-US with the linear matrix array probe ML6-15 (3DsnUS) and mechanically swept 3D-US with the automated motor array 3D convex probe RAB4-8 (3DmsUS) were applicable in all cases. With 3DmsUS, only 21 of the 31 patients were examined due to organizational reasons. However, in a few cases the image quality was not optimal due to artefacts or incompletely captured pols and isthmuses (Figure 7), but still sufficient for diagnostic demands. No examinations needed to be excluded because of poor image quality.

In cases of 3DsnUS, the examiner had to pay attention to a constant probe movement speed in order to avoid artefacts. From the patient’s side, these could be reduced by a breath-holding technique. Motion artefacts made the stitching process and the application of the MC method more difficult and could lead to measurement inaccuracies. In theory, electromagnetic interferences from metal, mains, or LAN voltage fields are possible, but has not been observed in our examinations.

For accurate 3DmsUS results, constant orthograde fixation of the probe on the neck surface was necessary. An important prerequisite was the estimation of the correct probe position in order to be able to capture the entire thyroid lobe, restricted by a maximum angle of 84°. Incompletely captured lobes needed to be avoided. In nearly all cases repetitive attempts very necessary. The acquisition parameters for 3D-US were individually optimized and, therefore, examiner-dependent, resulting in variable image quality.

The larynx was found to be an influencing factor in the acquisition process of 3D-US in general. In order to capture the thyroid at any height, the probe had to be moved around the larynx, resulting in a tilt of the 3D data sets. This was a challenging factor for the stitching process (Figure 8).

Digital Imaging and Communications in Medicine (DICOM) data exports to external storage devices as well as transferal to and post-processing in the research software PMOD and the clinical software Syngo.via were successfully performed in analogy to several previous studies [17,19,20]. For 3DmsUS, edge length voxel corrections were necessary (equally to previous study protocols) [20].

For stitching procedures, trachea, calcifications in the cartilaginous braces, and isthmus-related parenchymal changes such as nodules or cysts were found to be helpful anatomical landmarks. The larger the overlapping portions of both thyroid lobes, the higher the probability of detection of common landmarks. However, correct anatomical stitching is challenging. In the present study, fusion with the I-124-PET scans enabled accurate results but was a time-consuming process. In addition, variable head positioning during US and PET acquisitions could lead to variations in thyroid presentation. Overall, the manual stitching method was successfully performed in all investigated patients, but a major disadvantage was the relatively high time-consumption of the entire process ([Fig biomedicines-11-00381-ch003]). 

The methodological approach of US stitching has also been investigated in other disciplines, e.g., gynaecology, echocardiography, oncology, and ophthalmology [27,28,29,30]. A common goal is to optimize the image assessment with the help of an extended FOV, resulting in one comprehensive 3D-US data set of the area of interest (in this case the entire thyroid gland). Important advantages are the possibilities to export and archive the acquired 3D-US data sets, allowing for retrospective reviewing, post-processing, post-hoc volume analyses, and secondary fusion with other 3D modalities [17]. The stitched 3D-US data sets of the entire thyroid gland can theoretically be fused with any other 3D images, e.g., 99m-technetium-pertechnetate- or I-123-SPECT(/CT) data (Figure 9). In contrast to I-124-PET/CT (~6.5 mSv), 99m-technetium-pertechnetate- or I-123-SPECT scans can be performed immediately after the standard planar scintigraphy without additional radiation exposure [31]. This can be very helpful for the functional assessment of thyroid nodules in cases of unclear conventional diagnostic results. Previous studies addressed this aspect for 99m-technetium-pertechnetate SPECT/US and I-124-PET/US fusion imaging and demonstrated relevant diagnostic benefits. However, so far PET/US and SPECT/US fusion imaging have mainly been reported in the context of real-time applications, but not for a secondary fusion approach.

A major advantage of 3D-US of the thyroid gland is the ability to apply segmental organ contouring (the MC method). In contrast to EM, this approach allows for accurate volume determination, especially cases of enlarged and deformed thyroid glands such as those investigated in the present study. The results show very high correlation coefficients between 3DsnUS-MC and 3DmsUS-MC with the reference standard. In comparison to the volumes determined by conUS-EM, 3D-US revealed favourable results with relevantly lower dispersions, particularly for the larger thyroid glands. In analogy to other studies, an average underestimation of thyroid volume was observed for the conUS-EM method and relative differences were found in a range of 13–23%, which is consistent with the results of this study (mean: 24.1%) [25,32,33,34]. Analyses of the patients with the highest differences between conUS-EM and the gold standard (>40%; N = 7) in this study revealed two main reasons for the inaccurate conUS-EM results:in large thyroid glands, the maximum cranial–caudal diameter could not be measured correctly on conUS because the respective organ pols were not covered within the limited FOVsubstantial deviation of the organ shape from the estimated ellipsoid model due to nodules and cysts, especially in the isthmus.

Another measurement error may result from the correction factor of the EM. Various factors have been proposed and discussed in the literature [35]. Initially, 0.479 was suggested [24]. Other studies used π/6 (=0.524) [36]. In addition, several slightly higher or lower values have been tested [32,37]. In clinical practice, the rounded factor of 0.5 is commonly used. Therefore, we decided to use the widespread clinical standard value for comparative analyses in the present study.

Previous publications were predominantly concerned with 3DsnUS and without the stitching method. The present study is the first to report the use of two different 3D-US applications including stitching in patients with structural thyroid disease. A summary of the literature is given in Table 3.

Schlögl et al. (2001), Lyshchik et al. (2004), and Licht et al. (2014) examined in vivo thyroid glands in addition to phantoms [14,15,38]. The differences and standard deviations are similar to the results of the present study. The mean values are slightly lower than in the current results with respect to the 3DsnUS. Standard deviations are almost the same. However, the thyroid glands examined in the studies of Schlögl et al. (2001) and Lyshchik et al. (2004) had significantly lower volumes, and in Schlögl et al. (2001) data sets with poor image quality were excluded. In the present study, no data sets were excluded regardless of the presence of poor image quality or image artefacts. Additionally, 3DmsUS results from the above mentioned studies were also comparable, but the standard deviation was higher in the present study, which can be explained by the more complex organ structures and types.

### Limitations

With regard to the study design, several limitations must be admitted that may have influenced the results and their interpretation:The investigated patient group consisted of subjects who received further diagnostics due to unclear constellations of conventional diagnostics. Thus, the 3D-US data were collected from a complex pre-selected patient cohort that does not reflect the average population. In the present study, the focus was set to patients with structural thyroid diseases in order to investigate the value of the novel methodology with regard to this aspect.Due to the small number of cases, the results have limited reliability statistically. Furthermore, the sample sizes of patients with 3DsnUS and 3DmsUS were different; only 21 of the 31 patients received 3DmsUS.MC volume determinations were performed exclusively by one examiner. Accordingly, no statement regarding interobserver variability is possible. On the other hand, there was no bias due to different levels of experience.Intraobserver variability was not investigated because each measurement was only carried out once. Given a total of 5.183 manually drawn contours (MC analyses), no effort was made to repeat the measurements.

## 5. Conclusions and Outlook

In this work, it was shown that the accuracy of volumetric determination of 3D-US data sets of the thyroid gland is superior to those of conventional 2D-US and comparable to those of other 3D sectional imaging methods such as CT. Additionally, 3D-US techniques combine the advantage of comparable precision with overcoming the disadvantage of radiation exposure of CT and the limited availability of MRI. Due to the multitude of benefits, ultrasound in general is one of the most important diagnostic imaging tools in clinical and ambulatory patient care. Nevertheless, limitations arise for the establishment of 3D-US procedures in clinical practice. These mainly relate to time expenditure and the limited availability of additional software still required for volumetric calculations.

Promising prospects are artificial intelligence (AI) approaches, which already exist in several fields of medicine, medical imaging, and ultrasound, in particular [42,43,44]. To overcome the limitations of high time-consumption and examiner-dependency of the MC method, AI-based automated segmentation methods have been evaluated [45,46]. Chen et al. summarized the various methods already developed for automated segmentation of the thyroid gland from 2D and 3D data sets in 2020 [25,32,33,34,47,48]. However, all methods so far focused exclusively on normal thyroid glands, publications addressing automated segmentation of pathological or atypically shaped thyroid glands are still pending. Furthermore, limitations currently arise from elaborate training processes and the need for extensive data input [49].

## Data Availability

Due to ethical regulations, the data will not be made public.

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
