# Peer review of "Optimization of Thyroid Volume Determination by Stitched 3D-Ultrasound Data Sets in Patients with Structural Thyroid Disease"

_biomedicines, 2023, doi:10.3390/biomedicines11020381_

Round 1

Reviewer 1 Report

This paper challenges more accurate thyroid volume measurement by stiched 3D measurement using a cervical ultrasound system. Accurate determination of thyroid volume is essential in determining the efficacy of medication in Graves' disease and in determining the radiation dose for radioiodine therapy. It is very important to follow the size of goiter in chronic thyroiditis with diffuse goiter. It is also important to follow the size of the goiter before thyroid surgery. Until now, thyroid volume has been measured using devices such as CT and PET/CT, which could not be easily implemented from the standpoint of cost and radiation exposure.

However, the main drawback of this method is that it is very time consuming. Therefore, it will not be widely used in general. The main drawback of this method is that it is very time consuming. Therefore, it would be difficult to spread this method in general. I would appreciate it if you could provide a simpler method, even if it is slightly less accurate.

PET/CT images are used as a reference standard, but PET/CT images are not truly accurate. Most desirably, in patients undergoing total thyroidectomy, the actual thyroid volume removed should be compared to the weight measurements from stiched 3D US measurements immediately prior to surgery. If a collaborative study with the Department of Surgery is possible, it would be highly recommended that this be conducted.

Author Response

see word document

Reviewer 2 Report

General remarks

The paper by Philipp Seifert et al presents an interesting basic study that compares several available methods for measurement of thyroid volume. The novelty of the paper is primarily related to the so called stitching used to join 3D images of both lobes. The study was well described and presented conclusions are justified. I especially appreciate that the authors included a graphic showing the workflow of 3D-US stitching procedure including time consumed at each step. I have one general remark though: I agree with the authors that given high costs and a great deal of time consumed by any alternative method, the conventional US remains the only method of thyroid volume determination feasible in the clinical setting. Thus, it would be very interesting to look closer at those cases in which conUS-EM result deviated strongly from the standard values determined with PET/CT-MC. Maybe there was some identifiable source of the deviation that could be taken into account in clinical practice.

Specific remarks

Line 54: “de facto unlimited repeatability” is used to describe the nearly unlimited possibility to repeat the US examination, but the repeatability of a diagnostic procedure means something different – that the repeated measurement gives the same result.

Line 57: “An accurate US-based determination of the thyroid volume is crucial for treatment planning, outcome assessment and follow-up in the therapy of thyroid diseases” seems to be an overstatement. The word “crucial” is very strong here and in many thyroid diseases the assessment of thyroid volume is rather irrelevant. So maybe the sentence should be more specific or focus on 131I therapy used for some thyroid disorders, as mentioned in the sentence that follows. The same applies to the analogous statement in the Abstract.

Line 192: “A total number (N) of 31 patients (12 men, 19 women, age: 57±15 years) were included, 21 only received 3DsnUS but no 3DmsUS.” – This sentence implies that 3DmsUS was performed in 10 patients only, as 21 patients did not ‘receive’ 3DmsUS. (the verb ‘receive’ sounds a kind of strange here). But Table 2 and plot C in Graphic 2 show that 3DmsUS was done in 21 patients. Please correct the indicated sentence.

Lines 246-248: “Sensor-navigated 3D-US with the linear matrix array probe ML6-15 (3DsnUS) and mechanically-swept 3D-US with the automated motor array 3D convex probe RAB4-8 (3DmsUS) were applicable to all examined patients. However, in a few cases the image quality was poor due to artefacts or incompletely captured pols and isthmuses (Figure 7).” – should we understand from these sentences that 3DmsUS examination was performed in all 31 patients but in some cases the poor quality made it impossible to determine the thyroid volume. And that is the reason for 3DmsUS measurements in only 21 patients. If so, why the discussion is concluded with the following statement: “In the present study, no data sets were excluded regardless of the presence of poor image quality or image artefacts” (at line 351)?

conUS-EM gave on average lower volumes than other evaluated methods. Please comment why the simplified formula for ellipsoid volume was used (xyx * 0.5) instead of the exact one (xyz * π/6). I have noticed a short paragraph in the discussion related to this issue, but it seems quite obvious that 0.5 factor is used in clinical practice for its simplicity. The present study was extremely laborious comparing to the effort needed to use a more exact factor.

Author Response

see word document
